# Comorbidity and Overlaps between Autism Spectrum and Borderline Personality Disorder: State of the Art

**DOI:** 10.3390/brainsci13060862

**Published:** 2023-05-26

**Authors:** Liliana Dell’Osso, Ivan Mirko Cremone, Benedetta Nardi, Valeria Tognini, Lucrezia Castellani, Paola Perrone, Giulia Amatori, Barbara Carpita

**Affiliations:** Department of Clinical and Experimental Medicine, University of Pisa, Via Roma 67, 56127 Pisa, Italy; liliana.dellosso@gmail.com (L.D.); ivan.cremone@gmail.com (I.M.C.); valeria_tog@hotmail.com (V.T.); castellanilucrezia91@gmail.com (L.C.); p.perrone317@gmail.com (P.P.); giulia.amatori@libero.it (G.A.); barbara.carpita1986@gmail.com (B.C.)

**Keywords:** autism spectrum disorder, autistic traits, borderline personality disorder, personality disorders

## Abstract

Despite the relationship between Autism spectrum disorder (ASD) and personality disorders (PD) still being scarcely understood, recent investigations increased awareness about significant overlaps between some PD and autism spectrum conditions. In this framework, several studies suggested the presence of similarities between BPD and ASD symptoms and traits, based on the recent literature that increasingly reported increased comorbidity rates and significant symptomatologic overlaps between the two conditions. The aim of this review is to describe the available studies about the prevalence of the association between different forms of autism spectrum (full-fledged clinical conditions as well as subthreshold autistic traits) and BPD. Despite some controversial results and lack of homogeneity in the methods used for the diagnostic assessment, the reviewed literature highlighted how subjects with BPD reported higher scores on tests evaluating the presence of AT compared to a non-clinical population and hypothesized the presence of unrecognized ASD in some BPD patients or vice versa, while also describing a shared vulnerability towards traumatic events, and a greater risk of suicidality in BPD subjects with high autistic traits. However, the specific measure and nature of this association remain to be explored in more depth.

## 1. Introduction

Autism Spectrum Disorder (ASD) is a neurodevelopmental disorder that features behavioral patterns, restricted and repetitive interests and chronic difficulties in social communication and interaction [1]. Interestingly, a central feature of ASD that has recently gained more interest is an alteration in sensory processing that occurs when the ability to respond behaviorally to sensory information, including sound, touch, body movement, sight, taste and smell, is diminished. This alteration can ultimately lead to unusual responses to sensory inputs, which may impair the daily life activities of ASD subjects [2]. Even though ASD etiopathogenesis is still unclear [3], there is a good amount of evidence suggesting the relevance of both genetic correlates [4] and the environment [5,6]. The definition of ASD includes conditions with different levels of severity, with or without intellectual impairment or language development alterations. The DSM-5-TR recognizes three levels of severity: the first one includes deficits in social communication that, without support, cause noticeable impairment while the second level requires marked deficits in verbal and nonverbal social communication skills and great distress and/or difficulty changing focus or action. Lastly, the third level presents severe deficits in verbal and nonverbal social communication skills that cause severe impairments in functioning, very limited initiation of social interactions and minimal response to social overtures from others, as well as extreme difficulty coping with changes, or other restricted/repetitive behaviors which markedly interfere with functioning in all spheres [1]. Despite the fact that research on ASD has been mainly conducted on children, studies on this condition are equally significant for adults because it frequently co-occurs with other psychiatric disorders. The existence of undiagnosed ASD among inpatients seeking treatment is a topic of great therapeutic importance, emphasizing the necessity of rigorous research of autistic symptoms in clinical samples as well as in the general population [7].

A central feature of ASD is the presence of an atypical social approach to dialogue reciprocity and a decreased sharing of interests, emotions and sentiments, caused by difficulties in the development of an adequate Theory of mind (ToM) [7,8]. It should be noted that, in the last decades, research in the field of ASD stressed the need to not limit the investigation to full-blown clinical forms, but to also evaluate those milder, sub-clinical manifestations of the autism spectrum which seem to be distributed along a continuum from the general to the clinical population [9,10,11,12,13]. Sub-threshold autistic traits were first investigated among first-degree relatives of ASD patients, where they are known under the name of “broad autism phenotype” [14,15]. However, further studies identified other populations at higher risk of showing autistic traits, ranging from students of scientific courses to psychiatric patients with other kinds of disorders [3,15,16,17,18,19,20,21,22]. These traits are divided into various dimensions, although research indicates that not all of these dimensions strongly correlate with one another [23] and that different dimensions may be linked to various outcomes [24]. For example, Davis et al. [24] reported that the social and non-social facets of autistic traits differentially predicted social cognitive processes in a sample of college students. Additionally, the various facets of autism traits appear to be connected with sensory processing in adults with typical development in diverse ways [25,26]. The interest in focusing on subthreshold autistic traits lies in the fact that they seem to exert a detrimental effect on quality of life, being also a significant vulnerability factor for the development of other psychiatric disorders, as well as suicidal thoughts and behaviors [26,27,28]. As the autism spectrum is frequently closely associated with other features beyond its core characteristics, two of which are alexithymia [29,30] and anxiety [31,32], understanding the structure of autism traits and supporting their adequate measurement allows for the further assessment of the relationship between these traits and other variables of interest in the spectrum. In fact, high levels of anxiety [33,34] and anxiety disorders [32,35] are also often reported in youth and adults with autism, influencing their life outcomes [36]. Due to its pervasive, multiple and kaleidoscopic manifestations, which are often associated with anxiety and mood symptoms, recognizing high-functioning forms of ASD, and even more subthreshold autistic traits, among adults with other comorbid psychiatric disorders can sometimes be difficult, especially when a personality disorder (PD) is present or suspected [37,38,39,40,41].

According to the Diagnostic and Statistical Manual of Mental Disorders (DSM) [1] and the International Classification of Diseases (ICD) [42], a PD is a persistent, pervasive and rigid pattern of inner experience and conduct that diverges significantly from cultural norms and expectations, which can influence cognition, emotion, interpersonal functioning and impulse control. PD manifestations typically start in adolescence or early adulthood and are present at any age. For the diagnosis of PDs, it is necessary to assess the long-term functioning patterns of the person and the specific personality traits must be present by early adulthood. Additionally, the personality features typical of PDs must be separated from symptoms and traits that arise in reaction to certain situational stressors or from more ephemeral or episodic mental states (such as bipolar, depressive, or anxiety disorders, or drug intoxication) [1]. Three distinct clusters of PDs have been identified. The first group (Cluster A), sharing strange or unconventional appearance and traits, includes schizoid, schizotypal and paranoid PDs. Cluster B features instead narcissistic, borderline, histrionic and antisocial PDs, which frequently manifest in people as dramatic, emotional or erratic behavior. Lastly, obsessive-compulsive, dependent and avoidant PDs are all included under Cluster C.

Even though the relationship between ASD and PD is still scarcely understood, recent investigations increased awareness about some symptoms’ similarities, which makes differentiating diagnostic evaluation more difficult [39,40,41,43]. Moreover, considering that both ASD and PD feature persistent traits [1], methods used to distinguish between these two conditions, such as looking for changes in a patient’s level of functioning or behavior, could be scarcely effective [37]. According to a recent literature review, around 50% of subjects with ASD may also meet the diagnostic criteria for at least one PD [44]. In particular, some papers stressed how PDs seem to be a fairly common misdiagnosis among unrecognized ASD adults [45] while other authors highlighted a high rate of comorbidities between these two kinds of disorders [46,47,48]. Among PDs, borderline personality disorder (BPD) is one of the most investigated in the literature due to its higher prevalence with respect to other PDs as well as its associated clinical challenges [49].

As defined by the DSM-5-TR and the ICD-11 [1,42], BPD is characterized by a pervasive pattern of instability in affect regulation, impulse control [50,51], interpersonal functioning (such as altered empathy and problems with trust and intimacy) [40,52], unstable mood and self-image, emotional dysregulation and fear of abandonment [1] and difficult personality traits (such as disinhibition and antagonism). BPD has a lifetime prevalence of 5.9% and is more often diagnosed in females [53]. These patients frequently seek out mental-health services due to the clinical indications of the illness, which include emotional dysregulation, impulsive violence, repetitive self-injury and chronic suicidal thoughts. Although the causes of the disorder’s development are only partially understood, hereditary factors and negative childhood experiences, such as physical and sexual abuse, were reported to play a role [54]. Although the two disorders may seem quite different, recent literature increasingly reported significant overlaps. The two disorders mainly differentiate for the presence of restricted interests and repetitive behaviors, as well as an alteration in sensory processing that, while being necessary for the diagnosis of ASD, it is not required for that of BPD [1]. On the other hand, while BPD is characterized by a pervasive instability of relationship and self-image, feelings of emptiness and desperate efforts to avoid abandonment, those are not required features for a diagnosis of ASD [1]. Moreover, the two disorders often report different triggers of emotional upset. For example, in ASD emotional outbursts may be triggered by changes in the daily routine or by a cognitive or sensitive overload, while in BPD by attachment issues.

On the other hand, interestingly, several studies have suggested similarities between BPD and ASD symptoms and traits [50,51,52,53] (Figure 1). For example, intense relationships and superficial friendships, as well as the tendency of acting out instead of verbalizing emotions, even if they are typical features of BPD, are also common in ASD. ASD subjects also reported consistent rates of self-injurious behaviors, which is another feature shared with BPD [55]. Similarly, impairment in verbal and non-verbal communications, social functioning, erroneous assumptions about motives and emotional meltdowns [56,57], while being core features of ASD, are also frequently reported in BPD subjects [3,21,58,59,60,61,62,63]. Furthermore, a variety of studies have reported that not only could many BPD traits be considered a consequence of emotional dysregulation [64,65,66,67], a dimension also largely represented in ASD subjects [68], but also that people with BPD show difficulties in identifying, distinguishing and integrating their emotions with those of other individuals [69]. While impaired social and relational areas may be a common core for ASD and BPD [70], some authors stressed in BPD subjects difficulties in the ToM similar to those typical of ASD subjects [70,71,72,73,74]. This is particularly relevant since altered social perception and altered functioning of more sophisticated neurocognitive skills such as ToM and associated mentalizing (i.e., the capacity to predict another person’s conduct based on their mental state) may lead, in both BPD and ASD patients, to emotional disturbances, and impulsive and self-destructive behaviors [75,76]. Additionally, both BPD and ASD symptoms have been reported to improve after therapeutic interventions that target emotion recognition, mentalizing and empathy, such as ToM and mentalization-based psychotherapy [77,78] and oxytocin intranasal administration [79,80]. A possible explanation for this concordance could be that the neurocognitive aspects of ASD that are impaired, such as mentalizing, impulse control, empathy and communication, may have a major influence on how personality develops [81]. However, besides studies generically investigating the presence of psychopathological traits typical of one of these two conditions in the other [55,61,62,63,64,65,66,67,68,69,70,71,72,73,74], or case reports [82], the available literature specifically focused on investigating prevalence and correlations between the autism spectrum and BPD is still limited, and differences between studies assessing sub-threshold or full-threshold autism conditions should be clarified (Figure 1)

Although in the recent literature, an increasing number of studies are analyzing the possible overlaps and correlations between ASD and BPD, to date, the literature in this area is still fairly young, and more has to be understood about the specific features of these correlations, the nature of the symptomatologic overlaps and how they relate to the manifestation of both disorders and the overall psychopathological illness trajectory. To our knowledge, to date the topic has been assessed exclusively by one meta-analysis, but without including subthreshold autistic traits [83]. In this framework, the aim of this review was to describe the available studies about the prevalence of the association between different forms of autism spectrum, including subthreshold autistic traits, and BPD, focusing also on possible clinical correlates. In the discussion section, a dimensional hypothesis to the psychopathological trajectory is presented, which would start from a vulnerability represented by autistic traits and identifies traumatic events and BPD as further steps and crossroads for the development of psychopathology.

## 2. Association between ASD and BPD

### 2.1. BPD and ASD

One of the first studies aiming to examine the comorbidity between full-blown ASD and BPD came from Rydén et al. [43], who compared BPD patients with or without ASD with respect to significant variables such as suicide attempts, self-harm, inpatient days and symptom burden, and identified some distinctive traits found in individuals who have both ASD and BPD. The autistic features of BPD female patients were evaluated with interviews, neuropsychological tests and questionnaires as well as with the examination of medical records. ASD has been assessed using the diagnostic standards for Asperger’s syndrome, ASD and pervasive developmental disorder NOS. For adolescents, the Asperger Syndrome Diagnostic Interview (ASDI) was used alongside the “Five to fifteen” (FTF) for the autistics’ attention deficit hyperactivity disorder and other comorbidities (A-TAC). Memory, learning, language, executive functions, motor skills, perception, social skills and emotional/behavioral issues were all assessed with the FTF. Cut-off scores for neuropsychiatric disorders were provided by the ATAC interview. The results were discussed with a skilled clinician trained to evaluate and treat ASD and ADHD/ADD. The authors reported that six of 41 patients (15%) with SCID-II (Structured Clinical Interview for DSM-IV Axis II Disorders)—verified BPD met the criteria for ASD. BPD patients with ASD also attempted suicide more frequently, and showed a higher substance abuse and altered self-image, with even less self-love and more self-hate. Furthermore, based on the lower Global Assessment of Functioning (GAF) scores obtained by ASD subjects, the authors concluded that BPD patients with autistic traits had a higher disease severity. In general, the presence of comorbid ASD in BPD patients resulted in lower mentalizing capacity and lower social functioning that possibly leads to higher levels of anxiety due to interpersonal problems and a compromised self-image. In a retrospective case-control study by Shen et al. [84], 292 newly diagnosed BPD patients and 5840 controls selected from the National Health Insurance Research Database were compared to determine the prevalence of psychiatric comorbidity throughout a 3-year period before the BPD diagnosis. Among other comorbidities, they found in BPD patients an increased odds ratio of having an ASD diagnosis (OR = 10.0). In order to identify ASD in adults, Brugha et al. [85] compared the effectiveness of the Autism Quotient (AQ) and the Ritvo Autism-Asperger’s Diagnostic Scale-Revised (RAADS-R) questionnaires in adult mental health services in two English counties. Subsequently, they recruited 364 men and 374 women who previously completed the AQ and RAADS-R to undergo an assessment with the Autism Diagnostic Observation Schedule (ADOS Module 4). The authors identified more unrecognized ASD subjects among women than men. Among others, five of the women with ADOS-determined ASD diagnosis (45%) had a diagnosis of BPD: these data may also support the hypothesis of a greater misdiagnosis of ASD among women [43]. It can then be inferred that disorders such as BPD can mask the presence of ASD, especially in women, who are also more likely than men to repress any autistic traits by mimicking normal behavior. Moreover, Hermann et al. [86] conducted an observational study by analyzing diagnostic information taken from their BPD-specific ward and found that ASD was diagnosed in 2.7–5.7% of BPD patients.

Other authors investigated instead the prevalence of BPD among subjects with ASD. In a 2006 study, Anckarsäter et al. [81] evaluated 240 subjects (131 men and 109 women) for ASD and Attention Deficit Hyperactivity Disorder (ADHD) using the Temperament and Character Inventory (TCI), identifying axis II personality disorders in a subgroup of 174 subjects using the SCID-II. They found an ASD diagnosis in 113 subjects, an ADHD diagnosis in 147 subjects, while 27 subjects had neither ASD nor ADHD, but other diagnoses. All subjects were tested with the ASDI, the Asperger Syndrome and High-Functioning Autism Screening Questionnaire (AS-HFASQ), the Yale–Brown Obsessive Compulsive Scale (Y-BOCS), the Structured Clinical Interview for DSM-IV Axis I Disorders (SCID-I) and DSM-IV criteria were used to check for ASD, ADHD, Impulse Control Disorders, Tic Disorders and other relevant disorders. The authors found that both ASD and ADHD subjects exhibited higher rates of Personality Disorders. According to their results, Obsessive-compulsive Personality Disorder was more common in ASD subjects, and BPD more common in ADHD subjects. Regardless of prevalence, these results highlight how a neurodevelopmental disorder, particularly if not diagnosed at an early age, can be identified as a personality disorder if evaluated in adult patients. Hofvander et al. [87] showed that, in a sample of 62 high-functioning ASD individuals, 68% met the criteria for one or more personality disorders, 40% met those for two or more and 18% met those for three or more: in particular, 7% of the population had BPD. Similarly, Strunz et al. [88] tried to compare the features of PD and ASD without intellectual impairment. They enrolled 106 controls, 62 individuals with Narcissistic Personality Disorder (NPD), 80 individuals with BPD and 59 individuals with an ASD diagnosis (83% had Asperger syndrome and 17% had high-functioning autism). The ASD diagnosis was made through the ADOS, which is a clinical evaluation based on the DSM-IV diagnostic criteria and the Autism Diagnostic Interview-Revised (ADI-R). For determining the presence of axis I disorder in ASD and BPD patients was used the German version of the Mini International Neuropsychiatric Colloquium (M.I.N.I.), while the German version of the SCID-I according to DSM-IV was used specifically for NPD patients. All participants were evaluated with the NEO-Personality Inventory-Revised (NEO-PI-R), a self-report questionnaire used to evaluate the Five-Factor Model’s (FFM) five personality traits: openness to experiences, neuroticism, sympathy, extraversion and conscientiousness. The subjects were also tested with the Dimensional Assessment of Personality Pathology (DAPPBQ), which is a dimensional self-report questionnaire for measuring pathological personality traits. According to the study, there are significant differences between the pathological personality features of BPD and NPD patients, ASD patients and non-clinical controls. In particular, ASD subjects scored considerably higher on the DAPP-BQ component Emotional Dysregulation and the NEO-PI-R Neuroticism dimension than controls, while they scored lower in comparison to BPD patients. Compared to all other groups, ASD patients reported considerably lower scores on the NEO-PI-R Extraversion dimension. ASD patients also scored considerably higher on the DAPP-BQ dimension Inhibitedness than NPD patients and controls. There was no statistically significant difference between the ASD and BPD groups on the Inhibitedness dimension. The NEO-PI-R Openness to Experience dimension, particularly the subscales Aesthetics, Emotions and Actions, had the lowest scores among all groups in ASD patients. Furthermore, individuals with ASD scored much higher on the subscale Ideas than BPD patients but did not statistically differ from NPD patients or non-clinical controls. The Agreeableness scores of ASD patients were significantly lower than those of nonclinical controls, but there were no significant differences between the ASD group and individuals with BPD or NPD. Moreover, ASD patients scored higher than all other groups on the NEO-PI-R subscale for Straight-forwardness. In the DAPP-BQ scale dimension Dissocial Behaviour, ASD participants scored statistically similar to nonclinical controls and substantially lower than BPD and NPD patients. Considering the NEO-PI-R Conscientiousness dimension, ASD patients scored considerably higher than BPD and NPD individuals, while considering Compulsivity in the DAPP-BQ scale they scored significantly higher than all other groups. As compared to all other groups, ASD patients scored considerably higher on the NEO-PI-R Straightforwardness subscale, while they scored similar to nonclinical participants in Modesty and Compliance subscales. Lastly, they found that the DAPP-BQ Narcissism subscale results for ASD patients and nonclinical controls were similar. To conclude, what emerges from this study is that there are clear differences between the personality profiles of patients with BPD and NPD and patients with ASD. Subjects with ASD are open to exclusively intellectual experiences and show high levels of introversion in almost all areas. Compared to BPD patients specifically, they showed higher levels of conscientiousness and introversion. In a 2007 study, Ketelaars et al. [89] highlighted the possible differences in comorbid Axis I and Axis II disorders in ASD and non-ASD groups. The results indicated that, except for Psychotic Disorder NOS which was diagnosed in roughly 20% of the non-ASD group and not in the ASD group, there were no significant differences in the pattern of diagnoses between the ASD and non-ASD patients. In a 2021 review, May et al. [83] aimed to analyze all evidence exploring the overlap of clinical presentation and co-occurrence between ASD and BPD. They examined 1633 studies, but only included 19 of them, including cross-sectional, cohort, case control and uncontrolled studies, exploring both ASD and BPD prevalence and/or phenomenology. Of these, 12 studies had appropriate data for the meta-analysis. Seven of them investigated the prevalence of BPD in ASD, which ranged between 0% to 12%, with a pooled prevalence of BPD in individuals with ASD of 4%. However, two of these seven studies did not confirm both ASD and BPD diagnosis, confirming only BPD but not ASD. The remaining five of the 12 suitable studies assessed ASD prevalence in BPD, highlighting a prevalence range of 0–15%, with a pooled prevalence of ASD diagnosis in individuals with BPD of 3%. One of these five studies did not confirm either ASD or BPD diagnosis, and one study confirmed ASD diagnosis but not BPD. Excluding these two studies, the pooled prevalence of ASD in BPD based on the remaining three studies was 5%. A summary of the described studies is shown in Table 1.

### 2.2. BPD and Subthreshold Autistic Traits

In addition to the studies that investigated the comorbidity between full-blown ASD and BPD, other studies focused on evaluating the prevalence and correlations of sub-threshold autistic traits in patients with BPD. In the study by Nanchen et al. [90], 38 female participants with BPD were tested for the presence of autistic traits with the AQ, for their degree of empathy with the Interpersonal Reactivity Index (IRI) and for alexithymia with the Toronto Alexithymia Scale (TAS). According to the findings, about half of the BPD patients had rates above the ASD cut-off in the AQ test. This study also showed that cognitive empathy ratings were lower and alexithymia scores were higher in the subgroup with significant autistic features. In another work, Dudas et al. [91] recruited 624 individuals diagnosed with ASD, 23 patients with BPD, 16 patients with comorbid ASD and BPD and 2081 neurotypical controls. All participants completed the following self-administered questionnaires: the AQ for the assessment of autistic traits, the Empathy Quotient (EQ) for quantifying empathy and the Systemizing Quotient-Revised (SQ-R) for assessing the ability to systematize. Using the latter score, participants were categorized into one of five cognitive profiles. Results on the AQ score highlighted that patients with BPD showed higher levels of autistic traits than controls, although lower than the ASD group. However, the authors stressed that the difference between ASD and BPD group was only of little significance. Moreover, adults with ASD and co-existing BPD reported higher mean scores than adults with ASD alone, indicating that both disorders may have an additive influence on ASD trait scores. Regarding the EQ score, they found that the ASD group and the comorbid ASD + BPD group both performed worse than the BPD group, which was comparable to the controls. Lastly, they discovered that both the ASD and BPD groups outperformed controls on the SQ-R test. These findings suggest that BPD patients, as well as ASD patients, have a higher systemizing ability compared to controls. Considering that a high systematization is a typical feature of both disorders, they suggested a further interpretation of these results, as a compensatory mechanism to the high emotional instability that characterizes these disorders. All of this highlights the importance of identifying the presence of autistic traits in patients diagnosed with BPD, as some of them may have both disorders, especially patients with a history of childhood neglect or abuse. In a 2018 study, Dell’Osso et al. [21] examined 50 patients who had treatment between 2015 and 2016 and had a clinical diagnosis of BPD. The control group consisted of 69 healthy individuals without a history of mental illnesses. Participants were required to complete the Adult Autism Spectrum (AdAS Spectrum), the AQ and the Mood Spectrum Self Report Measure (MOODS-SR). Women made approximately 70% of the BPD group (35 out of 50 subjects with BPD diagnosis) and 60.9% (42 out of 69 healthy subjects) of the control group. According to this study, patients with BPD reported higher autistic scores than healthy controls. Using Structured Clinical Interview for DSM-5 Disorders (SCID-5), the authors evaluated the BPD group for lifetime comorbidities with other psychiatric conditions, finding that 27 subjects (54%) had a trauma-related disorder; moreover, 68% (34 subjects) of the BPD group had a history of physical or sexual abuse. Moreover, they found that two subjects (4%) suffered from Bulimia Nervosa (BN), one subject (2%) suffered from Anorexia Nervosa (AN), three subjects (6%) had Binge Eating Disorder (BED) and lastly two subjects (4%) had Other Feeding and Eating Disorders. Evaluating subjects with MOODS-SR, they also found that 34% of the BPD group (18 individuals) committed a suicide attempt. In their analysis, a moderate correlation was found between suicidality and the AdAS Spectrum total score, whereas a significantly higher score on the AdAS Spectrum was made by BPD subjects with history of physical or sexual abuse. These results can be interpreted in two ways. In the first place, patients with subthreshold ASD, just like those with full-blown forms, are more exposed to traumatic events, often being the target of sexual abuse, violence and bullying, and can therefore consequently develop BPD-like symptoms and post-traumatic stress symptoms. On the other hand, PTSD symptoms include feelings of detachment from others and decreased participation in or interest in meaningful activities. These may thus mimic typical symptoms of ASD and generate higher scores on questionnaires that investigate the autism spectrum. Chabrol et al. [92] studied instead borderline personality characteristics, autistic features, depressive symptoms and suicide ideation in 474 college students (95 males and 379 females) between the ages of 18 and 25 using self-administered questionnaires: the Personality Diagnostic Questionnaire-4 (PDQ-4) was used for examining borderline characteristics, the AQ was used for assessing autistic features, the Center for Epidemiologic Studies Depression scale (CES-D) for depressive symptoms and the three-item Garrison scale for suicidal ideation. Four groups of subjects emerged from the analysis of the results: High Traits subjects (individuals with high scores in both the autistic and borderline dimensions), Autistic Traits, Borderline Traits and Low Traits. The High Traits group and the Borderline Traits group had similar levels of depressive symptoms, but the former showed a higher level of suicidal ideation than the latter. After excluding the interference of risk factors for suicidality such as cannabis use or female sex, they concluded that autistic traits alone were not sufficient to explain the high levels of suicidal ideation of the High Trait group (the group with both autistic and borderline traits), which would be possibly ascribed to the interaction between borderline traits and autistic traits. In particular, the typical BPD trait of high emotional reactivity can interact with the feeling of despair and helplessness generated by the high sensitivity to stress typical of ASD, leading to suicidal ideation. Moreover, the social isolation that derives from the difficulty in interpersonal relationships typical of ASD constitutes a further suicidal risk factor. A more recent study [27] assessed the presence of autistic traits in a sample of 58 subjects with Bipolar Disorder (BD), 48 subjects with BPD diagnosis and 59 healthy controls, with a specific focus on which dimensions of the autism spectrum could represent predictive factors of suicidality. All subjects were assessed with the following self-report instruments: the AdAS Spectrum, the Ruminative Response Scale (RRS) and the MOODS-SR for examining mood symptoms and suicidality. This study highlighted that autistic traits and rumination were more represented in both BD and BPD groups than in control subjects. Both groups scored above the AdAS spectrum threshold for the presence of significant autistic traits. Moreover, the pattern of autistic traits was reported to be associated with suicidality: in particular, in the BPD group the AdAS Spectrum Non-verbal communication and Hyper/Hyporeactivity to sensory input dimensions were positive predictors of suicidality, while the Inflexibility and adherence to routine dimension seemed to be a negative predictor. A summary of the described research is shown in Table 2.

## 3. Discussion

In recent years, several studies have suggested a similarity between BPD and the manifestation of ASD without cognitive impairment [81,93]. The reviewed literature highlighted how subjects with BPD reported significantly higher scores on tests evaluating the presence of AT compared to a non-clinical population [22,28,91] while others hypothesized the presence of unrecognized ASD in some BPD patients or vice versa [43]. Despite some controversial results and lack of homogeneity in the methods used for the diagnostic assessment, the literature seems to globally point out an increased prevalence of ASD and subthreshold autistic traits in subjects with PD [87] and, specifically, with BPD, as well as an increased prevalence of BPD among subjects with ASD [83,90]. These data should be also considered in light of the recent literature which focused on possible causes of under-recognition of ASD among females [94,95], hypothesizing that females may undergo a greater societal pressure to conform, being more motivated to learn how to hide their autism (camouflaging) [95,96,97].

In particular, females with ASD may show a reduced impairment in social communication and interactions and be more aware of their social difficulties, thus developing higher social anxiety levels and more frequently adopting social camouflaging strategies in order to mask their difficulties [18,97,98,99]. This kind of coping strategy, while sometimes being socially advantageous, may cause many ASD females to go “under the radar” and so remain undiagnosed, besides implying a lot of mental fatigue, stress and increased anxiety and depressive symptoms [91,96,97]. Noticeably, another feature typical of ASD females was reported to be a different pattern of restrictive interests, which may include a specific focus on food and diet. On the basis of these considerations, and of the symptomatologic overlaps between ASD and Anorexia nervosa, including a common difficulty in Theory of Mind tasks, several authors hypothesized that Anorexia nervosa, and eventually other Feeding and Eating Disorders (FEDs), may be considered a female-specific ASD presentation [16,17,19,20,100]. While the first author to hypothesize a link between ASD and AN was Gillberg in the 1980s [101], to date several studies supported an association between autism spectrum and FEDs. Since Gillberg’s first idea, research on the connection between ASDs and eating disorders has advanced [101,102], suggesting that AN could be thought of as an empathy disease on the same spectrum of ASD, due to several parallels between the two conditions. Longitudinal studies that found how ASD was overrepresented in the AN community provided new information that helped to clarify this concept [16]. In this framework, two recent reviews from Dell’Osso et al. [16] and Carpita et al. [15] summarized the large amount of literature about the association of AN with both full-threshold and subthreshold autism spectrum, featuring both longitudinal and cross-sectional studies. According to the authors, which also provided insights on the historical development of the concept of AN up to the DSM-5, the reviewed evidence seems to support the hypothesis of a possible reconceptualization of FEDs in light of a neurodevelopmental approach. These data are of particular interest because FEDs also show a very high comorbidity with BPD, and both these conditions feature a higher prevalence among females. In this framework, the different presentations of ASD among females may definitely lead to a late diagnosis of ASD; some women may also be misdiagnosed with a condition with similar features but more frequently associated with the female gender, such as a FED or a BPD (which, in turn, often come together), because clinicians are not trained for searching the female presentation of ASD [103]. According to this hypothesis, the camouflaging behaviors typical of ASD females may also explain the dramatic, sometimes artefact-expressivity and communicative style often reported among subjects with BPD [22,97]. As a consequence, in light of the possible specific female presentations of ASD, the reported increased prevalence, with respect to general population, of autism spectrum conditions among BPD patients and of BPD among subjects with autism spectrum conditions, could still be an underestimation, the tip of a greater submerged iceberg: an issue that should be addressed by further research.

Another interesting focus of some research was the correlation between suicidality, ASD and BPD [56,92]. It is widely recognized that BPD subjects are at higher suicide risk compared to the general population [92,104,105], as well as the high prevalence of self-injurious behaviors in individuals with ASD [106,107]. In fact, one of the most unique characteristics of BPD is the chronic aspect of the suicidal ideation [108] and follow-back studies have found that suicide occurs in up to 10% of BPD cases [109,110]. Interestingly, studies examining the frequencies of diagnosis in subjects deceased by suicide via psychological autopsy and post-mortem interviews with families, PDs occurred in around half of the cases under the age of 35, and BPD was indeed the most common category [111,112,113]. Of particular interest is the report that near to a third of youth suicides, most of whom are male, can be diagnosed with BPD by psychological autopsy [111]. Accordingly, some studies highlighted a greater risk of suicidality in BPD subjects with high autistic traits [22,92], interestingly showing how not only a full-blown ASD enhanced the overall suicidality, but this was also greatly influenced by subthreshold autistic and borderline traits [21,28,51,92,114].

Another major link between ASD and BPD is represented by the shared vulnerability towards traumatic events, which can lead to higher rates of stress-related symptoms and altogether worsen the global clinical picture [115,116,117,118]. Even though the studies focusing on the correlation between high autistic traits and the likelihood of abuse are still scarce, a growing body of evidence is reporting how individuals with autism spectrum conditions often suffer from bullying, interpersonal trauma, violence and sexual abuse [118,119,120,121,122]. On the other hand, traumatic events such as episodes of abuse are also frequently reported by BPD patients [123,124,125] and, furthermore, higher levels of autistic traits have been reported in BPD subjects who suffered from abuse (either physical or sexual) than in those without a history of trauma [59]. A possible explanation of these data can be that subjects with high autistic traits (either picturing a full-blown or a subthreshold disorder) may not only face a higher risk of exposure to trauma, but also have an increased vulnerability to the effects of the event resulting in the manifestation of post-traumatic symptoms and BPD-like characteristics [55,126]. Noticeably, more vulnerable subjects, such as individuals in the autism spectrum, suffering from chronic exposure to traumatic events (even interpersonal) can develop a peculiar post-traumatic phenotype known as Complex PTSD (cPTSD) [43], distinguished by the presence of emotional liability, long-term instability interpersonal relationships, negative self-perception and maladaptive behaviors [38,116] that can be easily misdiagnosed as BPD [127], further enhancing the risk of a BPD diagnosis which would mask an underlying autism spectrum, especially among women.

## 4. Limits

This review should be considered in light of some limitations. First of all, this is a narrative review, so it lacks the systematicity and reproducibility of a systematic one. Secondly, the literature available is still limited and often based on small samples of subjects. Thirdly, the BPD being a disorder mainly represented in females, there could be a possible influence of gender in the symptomatologic manifestation, not yet investigated. Moreover, to this date there is a lack of validated and widespread instruments for the evaluation of BPD traits. Lastly, we recognize the presence of a recent meta-analysis on the topic that, however, did not include subthreshold autistic traits in the evaluation.

## 5. Conclusions

In conclusion, despite the limited literature available, some interesting correlations between BPD and ASD or autistic traits were highlighted. However, despite much evidence supporting the possible overlap between autism spectrum and BPD, the specific measure and nature of these associations remain to be explored. To date, many aspects of this dimension remain controversial and should be further investigated by longitudinal studies and integrated in light of a dimensional approach to psychopathology. Further studies in the field should focus on evaluating the overlapping features and comorbidity between autism spectrum and BPD in light of a gender-specific approach. Improving knowledge in this field may ultimately lead to an improvement in the available preventive strategies, diagnostic procedures and treatment options for these conditions, as well as to reach a better understanding of gender specific psychopathology.

## Figures and Tables

**Figure 1 brainsci-13-00862-f001:**
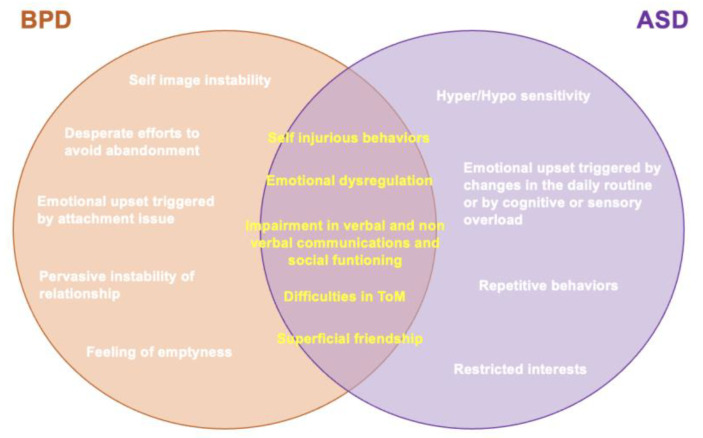
Differences and similarities of ASD and BPD.

**Table 1 brainsci-13-00862-t001:** BPD and ASD summary table.

Reference	Sample	Methods	Results
Rydén et al. 2008 [43]	BPD patients: F = 41	SCID-II; ASDI; FTF; A-TAC	15% of BPD subjects met the criteria for ASD More frequent suicide attempts, higher substance abuse and altered self-image in BPD subjects with ASD
Shen et al. 2016 [84]	BPD patients: N = 292 (F = 190; M = 102; mean age: 25 years) HC: N = 5840 (F = 3800; M = 2040; mean age: 25 years)	ICD-9-CM	increased odds ratio in BPD patients for having ASD (OR = 10.0)
Brugha et al. 2020 [85]	N = 378 (M = 364; F = 374)	AQ; RAADS-R; ADOS Module 4	45% of ASD female had a diagnosis of BPD
Anckarsäter et al. 2006 [81]	N = 240 (M = 131; F = 109; mean age: 31.0 years)	TCI; SCID-I; SCID-II; ASDI; AS-HFASQ; Y-BOCS	ASD and ADHD subjects exhibited high rates of Personality Disorders
Hofvander et al. 2009 [87]	ASD subjects: N = 62	ASDI; SCID-I; SCID-II	68% met the criteria for one or more personality disorders 40% met those for two or more, 18% met those for three or more: 7% of the population had BPD
Strunz et al. 2015 [88]	NPD: N = 62 (F = 17; M = 45; mean age 30.8 ± 9.7 years) HC: N = 106, 62 (F = 50; M = 56; mean age 32.7 ± 10.9 years) BPD: N = 80 (F = 51; M = 29; mean age 29.7 ± 8.8 years) ASD: N = 59 (F = 32; M = 48; mean age 32.7 ± 10.9 years)	ADOS; ADI-R; M.I.N.I.; SCID-I; NEO-PI-R; FFM; NEO-PI-R; FFM; DAPPBQ	DAPP-BQ: *Emotional Dysregulation scores:* BPD > ASD > HC *Inhibitedness scores:* ASD/BPD > HC *Dissocial Behavior scores:* BPD> ASD/HC *Compulsivity scores:* ASD > BPD/HC NEO-PI-R: *Neuroticism scores:* BPD > ASD > HC, *Extraversion dimension scores:* BPD/HC > ASD *Openness to Experience scores:* HC/BPD > ASD *Straight-forwardness scores:* ASD > BPD/HC *Conscientiousness scores:* ASD > PDB
Ketelaars et al. 2007 [89]	Outpatient: N = 369 (F = 189; M = 180; mean age: 35 years) ASD: N = 15 (F = 3; M = 12; mean age: 22 years) HC: N = 21 (F = 3; M = 18; mean age: 27 years)	ADI-R; ADOS; AQ	no significant differences in the pattern of diagnoses between ASD and non-ASD patients
May et al. 2021 [83]	12 studies		BPD prevalence in ASD ranged between 0% to 15% BPD pooled prevalence ASD individuals of 3–5%

**Table 2 brainsci-13-00862-t002:** BPD and subthreshold autistic traits summary table.

References	Sample	Methods	Results
Nanchen et al. 2016 [90]	BPD: F = 38	AQ; IRI; TAS	half of BPD subjects have above the ASD cut-off in the AQ test
Dudas et al. 2017 [91]	ASD: N = 624; BPD: N = 23; ASD + BPD: N = 16; HC: N = 2081; mean age: 39.43 ± 12.3 years	AQ; EQ; SQ-R	AQ score: ASD and ASD + BPD > BPD > HC EQ score: ASD and ASD + BPD performed worse than BPD (comparable to HC) SQ-R score: ASD and BPD >HC
Dell’Osso et al. 2018 [21]	BPD: N = 50 (F = 35; M = 15; mean age 33.8 ± 10.0 years) HC: N = 69 (F = 42; M = 27; mean age and 31.4 ± 11.4)	AdAS Spectrum; AQ; MOODS-SR; SCID-5	higher autistic scores in BPD subjects compared to HC higher AdAS Spectrum score in BPD subjects with history of physical or sexual abuse
Chabrol et al. 2018 [92]	N = 474 (F = 379, mean age: 20.7 ± 1.9; M = 95, mean age: 21 ± 2.3)	PDQ-4; AQ; CES-D; 3-item Garrison scale	similar levels of depressive symptoms in High Traits and Borderline Traits groups higher level of suicidal ideation in High Trait compared to Borderline Traits group autistic traits are not sufficient to explain the high levels of suicidal ideation of the High Trait group
Dell’Osso 2021 [27]	BD: N = 58 (F = 21; M = 37; mean age: 35.48 ± 11.24) BPD: N = 48 (F = 33; M = 15; mean age: 34.50 ± 9.67) HC: N = 59 (F = 32; M = 27; mean age: 32.86 ± 11.69)	AdAS Spectrum; RRS; MOODS-SR	autistic traits and rumination more represented in BD and BPD groups than HC BPD and BD scored above the AdAS spectrum threshold for the presence of significant autistic traits pattern of autistic traits was reported to be associated with suicidality

## Data Availability

All data generated or analyzed during this study are included in this published article.

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
