# Peer review of "Comorbidity and Overlaps between Autism Spectrum and Borderline Personality Disorder: State of the Art"

_brainsci, 2023, doi:10.3390/brainsci13060862_

Round 1

Reviewer 1 Report

Comorbidity and overlaps between autism spectrum and Borderline personality disorder: state of the art

The relationships between ASD and BPD are investigated through a review of the literature. The authors reviewed 8 studies that compared ASD as defined by meeting full criteria of the DSM with BPD and 5 studies that compared subthreshold ASD with BPD. Although there was little consistency in study methods and results, there appears to be note-worthy consistency in finding cross-diagnostic similarities in symptom presentation, personality traits, problematic behaviors, and social functioning. Specifically, the similarities are most notable when comparing subthreshold ASD and BPD which suggests that manifestations of ASD without cognitive impairment may be diagnostically more closely related to BPD. This is especially true for females. There are also high correlations of suicidality and trauma between the two groups. Future studies with a focus on clarifying the apparent diagnostic overlap between these disorders is needed to improve diagnosis and treatment of BPD and ASD, especially in females.

Introduction:

Adequate description of ASD and BPD. 

Association between ASD and BPD:

In general, this section could be improved with more interpretation of study results rather than the summary of results presented. This is not a necessary change, but could help to improve readability. Specifically, a significant portion of text includes the results of a study by Strunz et. al. (line 150-189) however, it is lacking a meaningful interpretation of the results. 

Tables:

Text in tables should not be justified.

Table 2 is split over multiple pages and is difficult to read.

BPD and subthreshold ASD: 

This is a good review of the literature and provides relevant information. However, if there is specific data within these studies about restrictive eating behaviors which is in the discussion, it would be important to highlight that data in this section. 

Discussion:

The paragraph that starts on 294 discusses results from studies that are not a part of this review or results that were not previously mentioned in the review such as the eating behaviors. This seems like relevant information that could be specified within the review.

General comments:

Poor translation - for example: repetitive use of the word “researches” instead of “studies”

Some of the information is difficult to understand due to the quality of the translation. Some words are not common in spoken English (anamnesis).

Author Response

Reviewer 1

The relationships between ASD and BPD are investigated through a review of the literature. The authors reviewed 8 studies that compared ASD as defined by meeting full criteria of the DSM with BPD and 5 studies that compared subthreshold ASD with BPD. Although there was little consistency in study methods and results, there appears to be note-worthy consistency in finding cross-diagnostic similarities in symptom presentation, personality traits, problematic behaviors, and social functioning. Specifically, the similarities are most notable when comparing subthreshold ASD and BPD which suggests that manifestations of ASD without cognitive impairment may be diagnostically more closely related to BPD. This is especially true for females. There are also high correlations of suicidality and trauma between the two groups. Future studies with a focus on clarifying the apparent diagnostic overlap between these disorders is needed to improve diagnosis and treatment of BPD and ASD, especially in females.

We thank the reviewer for giving us the opportunity to improve our work.

Introduction:

Adequate description of ASD and BPD. 

Response: we thank the reviewer for the comment. We also provided a more detailed description of ASD and BPD as requested by reviewer 3 (please refer to the responses to the reviewer 3 for more details).

Association between ASD and BPD:

In general, this section could be improved with more interpretation of study results rather than the summary of results presented. This is not a necessary change, but could help to improve readability. Specifically, a significant portion of text includes the results of a study by Strunz et. al. (line 150-189) however, it is lacking a meaningful interpretation of the results.                     

Thank you, we proceeded to add more comments in different points of the paragraph.

Tables:

Text in tables should not be justified.

Response: we thank the reviewer for the suggestion. We proceeded to properly format the text in the table.                                                                                                                                                                

Table 2 is split over multiple pages and is difficult to read.

Response: we thank the reviewer for the comment. We proceeded combine the table on one page to make it easier to read.

BPD and subthreshold ASD: 

 This is a good review of the literature and provides relevant information. However, if there is specific data within these studies about restrictive eating behaviors which is in the discussion, it would be important to highlight that data in this section. 

Discussion:

The paragraph that starts on 294 discusses results from studies that are not a part of this review or results that were not previously mentioned in the review such as the eating behaviors. This seems like relevant information that could be specified within the review.

We thank the reviewer for this comment. Actually, while to the best of our knowledge no study specific investigated the relationship between autism spectrum, BPD and eating behaviors all together, the literature about autism spectrum and eating disorders which we mentioned in the discussion section is quite large and reviewing it herein would overcome the aims of this work. In fact, the association between autism spectrum and eating disorders was already object of more than one review work from our group. As a consequence, following the suggestion of the reviewer, we provided more data about this subject but only in the discussion section, referring to our previous reviews, as follows:  

“While the first author to hypothesize a link between ASD and AN was Gillberg in the ‘80s [92]to date several studies supported an association between autism spectrum and FED. Since Gillberg's first idea, research on the connection between ASDs and eating disorders has advanced [92, 93], suggesting that AN could be thought as an empathy disease on the same spectrum of ASD, due to several parallels between the two conditions. Longitudinal studies that found how ASD was overrepresented in the AN community, provided new information that helped to clarify this concept [15]. In this framework, two recent reviews from Dell’Osso et al. [15] and Carpita et al. [14] summarized the large amount of literature about the association of AN with both full-threshold and subthreshold autism spectrum, featuring both longitudinal and cros-sectional studies. According to the authors, which also provided insights on the historical development of the concept of AN up to the DSM-5, the reviewed evidences seem to support the hypothesis of a possible reconceptualization of FED in light of a neurodevelopmental approach.”

General comments:

Poor translation - for example: repetitive use of the word “researches” instead of “studies”

Some of the information is difficult to understand due to the quality of the translation. Some words are not common in spoken English (anamnesis).

Response: we thank the reviewer for the comment, we proceeded to submit the text to proof reading.

Reviewer 2 Report

The present review bring a topic of autism spectrum disorder that describe the available researches about the prevalence of the association between 95 different forms of autism spectrum and BPD, focusing also on possible clinical correlates. I am giving several comments to the authors in the submitted manuscript.

1.      The present abstract was insufficient, please include the abstract's "take-home" message.

2.      Rearrange the keywords so that they are in alphabetical order.

3.      The novel of the present submitted article is not clear. Many published literature has been widely studied in the past. Further explanation in the introduction section in advance is mandatory.

4.      In line 21-42, related discussion of autism spectrum disorder, the authors encouraged to giving additional related literature as follows, doi: 10.3390/bioengineering9040157

5.      In order to demonstrate the work gaps that the current study aims to address, previous studies linked to it need to be explained in the introduction part, including their work, their novelty, and their limitations.

6.      It is suggested to the authors to make the objective of the present work become more clear to understand.

7.      Recommended to add an additional figure in the introduction section to improve the presentation of the present article.

8.      A comparative assessment with similar previous research is required.

9.      Because the current quality is not appropriate, the authors must improve their discussion to add more depth.

-

Author Response

Reviewer 2

The present review bring a topic of autism spectrum disorder that describe the available researches about the prevalence of the association between different forms of autism spectrum and BPD, focusing also on possible clinical correlates. I am giving several comments to the authors in the submitted manuscript.

We thank the reviewer for giving us the opportunity to improve our work.

  1. The present abstract was insufficient, please include the abstract's "take-home" message.

Response: we thank the reviewer for the comment. Upon the suggestion we decided to add some take-home messages in the Abstract as follows:

“Despite some controversial results and lack of homogeneity in the methods used for the diagnostic assessment, the reviewed literature highlighted how subjects with BPD reported higher scores on tests evaluating the presence of AT compared to a non-clinical population and hypothesized the presence of unrecognized ASD in some BPD patients or viceversa,  while also describing a shared vulnerability towards traumatic events, and a greater risk of suicidality in BPD subjects with high autistic traits.”

  1. Rearrange the keywords so that they are in alphabetical order.

Response: we thank the reviewer for the suggestion. We rearranged the keywords in alphabetical order.

  1. The novel of the present submitted article is not clear. Many published literature has been widely studied in the past. Further explanation in the introduction section in advance is mandatory.

Response: we thank the reviewer for the comment. The aim of our study was to be a review of the available literature on comorbidity and overlaps between ASD and autistic traits and BPD. To our knowledge, to this date a similar topic has been assessed exclusively by one meta-analysis. However, the presence of autistic traits and the related consequences, which are included in our review, were not analyzed in the previous meta-analysis. Moreover, our study embraces a dimensional approach to the psychopathological trajectory, starting from a vulnerability represented by autistic traits and proposing traumatic events and BPD as further steps and crossroads for the development of psychopathology.

Upon the suggestion, we decided to better explain in the Introduction as follows:

“To this date the topic has been assessed exclusively by one meta-analysis, but without including subthreshold autistic traits. In this framework, the aim of this review was to describe the available researches about the prevalence of the association between different forms of autism spectrum, including subthreshold autistic traits, and BPD, focusing also on possible clinical correlates. In the discussion section, a dimensional hypothesis to the psychopathological trajectory is presented, which would start from a vulnerability represented by autistic traits and identifies traumatic events and BPD as further steps and crossroads for the development of psychopathology.”

  1. In line 21-42, related discussion of autism spectrum disorder, the authors encouraged to giving additional related literature as follows, doi: 10.3390/bioengineering9040157

Response: we thank the reviewer for the comment. Following the suggestion, we added a short paragraph in the Introduction as follows:     

“Interestingly, a central feature of ASD that has recently gained more interest is an alteration in sensory processing, that occurs when the ability to respond behaviorally to sensory information, including sound, touch, body movement, sight, taste, and smell, is diminished. This alteration can ultimately lead to unusual responses to sensory inputs, which may impair the daily life activities of ASD subjects.”

  1. In order to demonstrate the work gaps that the current study aims to address, previous studies linked to it need to be explained in the introduction part, including their work, their novelty, and their limitations.

Response: we thank the reviewer for the comment. To this date, the literature in this area is still fairly young, and more has to be understood about the specifics of these correlation, the nature for the symptomatologic overlaps and how they relate to the manifestation of both disorder and the overall psychopathological illness trajectory. To our knowledge, only one meta-analysis has evaluated an issue like this so far. However, the meta-analysis did not examine the existence of subthreshold autistic traits nor the repercussions that followed; these topics are covered in our review. Additionally, our work adopts a multidimensional perspective on the psychopathological pathway, beginning with an autistic vulnerability and suggesting BPD as a turning point for the emergence of psychopathology. Upon the suggestion, we decided to better explain in the Introduction as follows:

“Although in the recent literature, an increasing number of studies are analyzing the possible overlaps and correlations between ASD and BPD, to this date, the literature in this area is still fairly young, and more has to be understood about the specific features of these correlations, the nature of the symptomatologic overlaps and how they relate to the manifestation of both disorders and the overall psychopathological illness trajectory. To our knowledge, to this date the topic has been assessed exclusively by one meta-analysis, but without including subthreshold autistic traits. In this framework, the aim of this review was to describe the available researches about the prevalence of the association between different forms of autism spectrum, including subthreshold autistic traits, and BPD, focusing also on possible clinical correlates. In the discussion section, a dimensional hypothesis to the psychopathological trajectory is presented, which would start from a vulnerability represented by autistic traits and identifies traumatic events and BPD as further steps and crossroads for the development of psychopathology.”

  1. It is suggested to the authors to make the objective of the present work become more clear to understand.

Response: we thank the reviewer for the comment. We proceeded to make the aims of the study more clear, changing the text as follows:

“In this framework, the aim of this review was to describe the available researches about the prevalence of the association between different forms of autism spectrum, including subthreshold autistic traits, and BPD, focusing also on possible clinical correlates. In the discussion section, a dimensional hypothesis to the psychopathological trajectory is presented, which would start from a vulnerability represented by autistic traits and identifies traumatic events and BPD as further steps and crossroads for the development of psychopathology.”

  1. Recommended to add an additional figure in the introduction section to improve the presentation of the present article.

Response: we thank the reviewer. Upon the suggestion, we also decided to add a Venn Graph to visually represent the differences and similarities of ASD and BPD, in order to improve the presentation.

  1. A comparative assessment with similar previous research is required.

Response: we thank the reviewer for the comment. The aim of our study is to be a review of the available literature on comorbidity and overlaps between ASD and autistic traits and BPD; therefore, not being a study with a sample, methods and results, it is not possible to make a comparison with other similar researches if not with previous reviews that have dealt with the same topic. To our knowledge, to this date a similar topic has been assessed exclusively by one meta-analysis [A1] however, the presence of autistic traits and the related consequences were not analyzed, which are instead included in our review. Moreover, our study embrace a dimensional approach to the psychopathological trajectory, which starting from a vulnerability represented by autistic traits and proposes BPD as a crossroads for the development of psychopathology.

We addressed the issue in the text as follows:

“To our knowledge, to this date the topic has been assessed exclusively by one meta-analysis, but without including subthreshold autistic traits. In this framework, the aim of this review was to describe the available researches about the prevalence of the association between different forms of autism spectrum, including subthreshold autistic traits, and BPD, focusing also on possible clinical correlates […]”

  1. Because the current quality is not appropriate, the authors must improve their discussion to add more depth.

Response: we thank the reviewer for the suggestion: as also requested by Reviewer 1, we added some missing elements in the discussion as follows:

““While the first author to hypothesize a link between ASD and AN was Gillberg in the ‘80s [92], to date several studies supported an association between autism spectrum and FED. Since Gillberg's first idea, research on the connection between ASDs and eating disorders has advanced [92, 93], suggesting that AN could be thought as an empathy disease on the same spectrum of ASD, due to several parallels between the two conditions. Longitudinal studies that found how ASD was overrepresented in the AN community, provided new information that helped to clarify this concept [15]. In this framework, two recent reviews from Dell’Osso et al. [15] and Carpita et al. [14] summarized the large amount of literature about the association of AN with both full-threshold and subthreshold autism spectrum, featuring both longitudinal and cros-sectional studies. According to the authors, which also provided insights on the historical development of the concept of AN up to the DSM-5, the reviewed evidences seem to support the hypothesis of a possible reconceptualization of FED in light of a neurodevelopmental approach.”

Reviewer 3 Report

Major:

Please sate the references list.

The logic of introduction is not clear. I'm not sure what the author is trying to state. The author should describe the ASD and BPD, separately. Next, state the differences and correlation, separately.

What is the classification g the PD? What’s the differences between PD and BPD?

Line 60-63: The references should be cited in the end of sentences.

Line 65-66: The author said the two disorders are quite different, what is the differences? And what is overlapped? If possible,a venn graph could help the reader to have a better understanding.

The paragraph 3 in Introduction is too long, please separate the similarity and differences in two paragraph.

What is the classification of ASD?

The logic of the part two is not clear. The paragraph is too long.

Minor:

1.       Line 54:What is PD28?Is there any mistake here?

2.       Line 59:The same problem. What is adults29?

3.       Line 64:  What is abandonment1?

4.       Line109:wrong pronunciation “autism – tics”

5.       Line 210:wrong subtitle number

English very difficult to understand/incomprehensible

Author Response

Reviewer 3

Major:

Please sate the references list.

Response: we thank the reviewer for the comment. The reference list is at the end of our paper and we also added more references.

The logic of introduction is not clear. I'm not sure what the author is trying to state. The author should describe the ASD and BPD, separately. Next, state the differences and correlation, separately.

Response: we thank the reviewer for the comment. We proceeded to describe ASD and BPD in separately as follows:

“Autism Spectrum Disorder (ASD) is a neurodevelopmental disorder that features behavioral patterns, restricted and repetitive interests and chronic difficulties in social communication and interaction [1]. Interestingly, a central feature of ASD that has recently gained more interest is an alteration in sensory processing, that occurs when the ability to respond behaviorally to sensory information, including sound, touch, body movement, sight, taste, and smell, is diminished. This alteration can ultimately lead to unusual responses to sensory inputs, which may impair the daily life activities of ASD subjects [2]. Even though ASD etiopathogenesis is still unclear [3], there is a good amount of evidences suggesting the relevance of both genetic correlates [4] and the environment [5, 6]. Despite the fact that research on ASD has been mainly conducted on children, studies on this condition are equally significant for adults because it frequently co-occurs with other psychiatric disorders. The existence of undiagnosed ASD among inpatients seeking treatment is a topic of great therapeutic importance, emphasizing the necessity of rigorous research of autistic symptoms in clinical samples as well as in the general population [7].”

And:

“As defined by the DSM-5-TR and the ICD-11, BPD is characterized by a pervasive pattern of instability in affect regulation, impulse control [36, 37], interpersonal functioning (such as altered empathy, and problems with trust and intimacy) [35], unstable mood and self-image, emotional dysregulation, fear of abandonment [1] and difficult personality traits (like disinhibition and antagonism). BPD has a lifetime prevalence of 5.9% and is more often diagnosed in females [34]. These patients frequently seek out mental-health services due to the clinical indications of the illness, which include emotional dysregulation, impulsive violence, repetitive self-injury, and chronic suicidal thoughts. Although the causes of the disorder's development are only partially understood, hereditary factors and negative childhood experiences, such as physical and sexual abuse, were reported to play a role [A8].”

As suggested, we then proceeded to list the differences and similarities the separately, adding a paragraph assessing the differences as follows:

“Despite the two disorders may seem quite different, recent literature increasingly reported significant overlaps. The two disorders mainly differentiate for the presence restricted interests and repetitive behaviors, as well as an alteration in sensory processing that, while being necessary for the diagnosis of ASD, it is not required for that of BPD [1]. On the other hand, while BPD is characterized by a pervasive instability of relationship and self-image, feelings of emptiness and desperate efforts to avoid abandonment, those are not required features for a diagnosis of ASD [1]. Moreover, the two disorders often report different triggers of emotional upset, for example, in ASD emotional outbursts may be triggered by changes in the daily routine or by a cognitive or sensitive overload, while in BPD by attachment issues.

On the other hand, interestingly, several researches have suggested similarities between BPD and ASD symptoms and traits [38-41]. […]”

What is the classification g the PD? What’s the differences between PD and BPD?

Response: we thank the reviewer for the question. We decided to better define a PD and its classification in the text as follows:

“According to the Diagnostic and Statistical Manual of Mental Disorders (DSM) [1] and the International Classification of Diseases (ICD) [26], a PD is a persistent, pervasive and rigid pattern of inner experience and conduct that diverges significantly from cultural norms and expectations, that can influence cognition, emotion, interpersonal functioning, and impulse control. PD manifestations typically start in adolescence or early adulthood, and are present at any age. For the diagnosis of PDs, it is necessary to assess the long-term functioning patterns of the person and the specific personality traits must be present by early adulthood. Additionally, the personality features typical of PDs must be separated from symptoms and traits that arise in reaction to certain situational stressors or from more ephemeral or episodic mental states (such as bipolar, depressive, or anxiety disorders, or drug intoxication) [1]. Three distinct clusters of PDs have been identified. The first group (Cluster A), sharing strange or unconventional appearance and traits, includes schizoid, schizotypal, and paranoid PDs. Cluster B features instead narcissistic, borderline, histrionic, and antisocial PDs, which frequently manifest in people as dramatic, emotional, or erratic behavior. Lastly, obsessive-compulsive, dependent, and avoidant PDs are all included under Cluster C.”

Line 60-63: The references should be cited in the end of sentences. 

Response: we thank the reviewer for the suggestion, however we believe that for a greater accessibility and comprehension, it is best to split the references in that paragraph, since each of them represents the source of each previous statement.

Line 65-66: The author said the two disorders are quite different, what is the differences? And what is overlapped? If possiblea venn graph could help the reader to have a better understanding.

Response: we thank the reviewer for the questions. We proceeded to list the differences and similarities the separately, adding a paragraph assessing the differences as follows:

“Despite the two disorders may seem quite different, recent literature increasingly reported significant overlaps. The two disorders mainly differentiate for the presence restricted interests and repetitive behaviors, as well as an alteration in sensory processing that, while being necessary for the diagnosis of ASD, it is not required for that of BPD [1]. On the other hand, while BPD is characterized by a pervasive instability of relationship and self-image, feelings of emptiness and desperate efforts to avoid abandonment, those are not required features for a diagnosis of ASD [1]. Moreover, the two disorders often report different triggers of emotional upset, for example, in ASD emotional outbursts may be triggered by changes in the daily routine or by a cognitive or sensitive overload, while in BPD by attachment issues.

On the other hand, interestingly, several researches have suggested similarities between BPD and ASD symptoms and traits [38-41]. […]”

Upon the suggestion, we also decided to add a Venn Graph to visually represent the differences and similarities.

The paragraph 3 in Introduction is too long, please separate the similarity and differences in two paragraphs.

Response: we thank the reviewer for the suggestion. We proceeded to divide the paragraph in smaller parts.

What is the classification of ASD?

Response: we thank the reviewer for the question. We added a paragraph in the text assessing ASD classification as follows:

“The definition of ASD includes conditions with different levels of severity, with or without intellectual impairment or language development alterations. The DSM-5-TR recognizes 3 levels of severity: the first one includes deficits in social communication that without support cause noticeable impairment while the second level requires marked deficits in verbal and nonverbal social communication skills and great distress and/or difficulty changing focus or action. Lastly, the third level present severe deficits in verbal and nonverbal social communication skills that cause severe impairments in functioning, very limited initiation of social interactions, and minimal response to social overtures from others as well as extreme difficulty coping with changes, or other restricted/repetitive behaviors which markedly interfere with functioning in all spheres [1].”

The logic of the part two is not clear. The paragraph is too long.

Response: we thank the reviewer for the suggestion. We proceeded to divide part two in smaller paragraphs.

Minor:

  1. Line 54:What is PD28?Is there any mistake here?

Response: we thank the reviewer for noticing. We provided to correct the mistake.

  1. Line 59:The same problem. What is adults29?

Response: we thank the reviewer for noticing. We provided to correct the mistake.

  1. Line 64:  What is abandonment1?

Response: we thank the reviewer for noticing. We provided to correct the mistake.

  1. Line109:wrong pronunciation “autism – tics” 

Response: we thank the reviewer for noticing. We provided to correct the mistake.

  1. Line 210:wrong subtitle number

Response: we thank the reviewer for noticing. We provided to correct the mistake.

Round 2

Reviewer 1 Report

This version is significantly improved. My previous comments were addressed appropriately. 

The grammar and translation are greatly improved!

Author Response

We thank the reviewer for the opportunity to improve our work.

Reviewer 2 Report

Following comments as the response in the present form given.

1.      What are the limitations of the current work? Please include it before the concluding section.

2.      Pease giving the explanation of correlation between autism spectrum disorder and anxiety behaviour. For this purpose, please refer the relevant reference as follows, doi: 10.3390/bioengineering9020048

3.      Further research should indeed be mentioned in the conclusion section.

4.      Throughout the manuscript, the authors created paragraphs that were only one or two phrases long, making the explanation difficult to understand. The authors should expand on their explanation to make it a more thorough paragraph. It is advised that one paragraph have at least three sentences, with one sentence functioning as the primary sentence and the other sentences functioning as supporting sentences.

5.      The authors need to enrich the reference from five years back.

6.      The authors were encouraged to proofread their work due to grammatical problems and linguistic style.

7.      The graphical abstract should be provided in the system after modification of peer review.

Following comments as the response in the present form given.

1.      What are the limitations of the current work? Please include it before the concluding section.

2.      Pease giving the explanation of correlation between autism spectrum disorder and anxiety behaviour. For this purpose, please refer the relevant reference as follows, doi: 10.3390/bioengineering9020048

3.      Further research should indeed be mentioned in the conclusion section.

4.      Throughout the manuscript, the authors created paragraphs that were only one or two phrases long, making the explanation difficult to understand. The authors should expand on their explanation to make it a more thorough paragraph. It is advised that one paragraph have at least three sentences, with one sentence functioning as the primary sentence and the other sentences functioning as supporting sentences.

5.      The authors need to enrich the reference from five years back. MDPI reference is strongly recommended.

6.      The authors were encouraged to proofread their work due to grammatical problems and linguistic style.

7.      The graphical abstract should be provided in the system after modification of peer review.

Author Response

Following comments as the response in the present form given.

We thank the reviewer for the opportunity to improve our work.

  1. What are the limitations of the current work? Please include it before the concluding section.

Response: we thank the reviewer for the question. We proceeded to add a paragraph before the conclusions, assessing the limitations of the current work as follows:

“This review should be considered in light of some limitation. First of all, this is a narrative review so it lacks of the systematicity and reproducibility of a systematic one. Secondly, the literature available is still limited and often based on small sample of subjects. Thirdly, being the BPD a disorder mainly represented in females, there could be a possible influence of gender in the symptomatologic manifestation, not yet investigated. Moreover, to this date there is a lack of validated and widespread instrument for the evaluation of BPD traits. Lastly, we recognize the presence of a recent meta-analysis on the topic that however, did not include subthreshold autistic traits in the evaluation.”

  1. Pease giving the explanation of correlation between autism spectrum disorder and anxiety behaviour. For this purpose, please refer the relevant reference as follows, doi: 10.3390/bioengineering9020048

Response: we thank the reviewer for the comment. Upon the suggestion we decided to add a paragraph in the Introduction assessing the correlation between autism spectrum disorder and anxiety, including the suggested reference, as follows:

“These traits are divided into various dimensions, although research indicates that not all of these dimensions strongly correlate with one another and that different dimensions may be linked to various outcomes. For example, Davis et al. reported that the social and non-social facets of autistic traits differentially predicted social cognitive processes in a sample of college students. Additionally, the various facets of autism traits appear to be connected with sensory processing in adults with typical development in diverse ways. The interest in focusing on subthreshold autistic traits lies in the fact that they seem to exert a detrimental effect on quality of life, being also a significant vulnerability factor for the development of other psychiatric disorders, as well as suicidal thoughts and behaviors. As the autism spectrum is frequently closely associated with other features beyond its core characteristics, two of which are alexithymia and anxiety, understanding the structure of autism traits and supporting their adequate measurement allows for the further assessment of the relationship between these traits and other variables of interest in the spectrum. In fact, high levels of anxiety and anxiety disorders are also often reported in youth and adults with autism, influencing their life outcomes.”

  1. Further research should indeed be mentioned in the conclusion section.

Response: we thank the reviewer for the suggestion. We proceeded to modify the conclusions including a mention on further researches ad follows:

“In conclusion, despite the limited literature available, some interesting correlations between BPD and ASD or autistic traits were highlighted. However, despite many ev-idences are supporting the possible overlap between autism spectrum and BPD, the specific measure and nature of these association remain to be explored. To this date, many aspects of this dimension remain controversial and should be further investigated by longitudinal studies and integrated in light of a dimensional approach to psycho-pathology. Further studies in the field should focus on evaluating the overlapping fea-tures and comorbidity between autism spectrum and BPD in light of a gender specific approach. Improving knowledge in this field may ultimately lead to an improvement on the available preventive strategies, diagnostic procedures and treatment options for these conditions, as well as to reach a better understanding of gender specific psychopathology.”

  1. Throughout the manuscript, the authors created paragraphs that were only one or two phrases long, making the explanation difficult to understand. The authors should expand on their explanation to make it a more thorough paragraph. It is advised that one paragraph have at least three sentences, with one sentence functioning as the primary sentence and the other sentences functioning as supporting sentences.

Response: we thank the reviewer for the comment. We controlled thoroughly the manuscript to make sure that every paragraph was formed at least by three of more sentences.

  1. The authors need to enrich the reference from five years back. 

Response: we thank the reviewer for the comment. We proceeded to upgrade our reference list including works from five years back.

  1. The authors were encouraged to proofread their work due to grammatical problems and linguistic style.

Response: we thank the reviewer for the comment, we proceeded to submit the text to proof reading.

  1. The graphical abstract should be provided in the system after modification of peer review.

Response: we thank the reviewer for the suggestion, we provided a graphical abstract.

Reviewer 3 Report

none

none

Author Response

(The authors gave the same response as above.)
